# Correction Model for Metal Oxide Sensor Drift Caused by Ambient Temperature and Humidity

**DOI:** 10.3390/s22093301

**Published:** 2022-04-26

**Authors:** Abdulnasser Nabil Abdullah, Kamarulzaman Kamarudin, Latifah Munirah Kamarudin, Abdul Hamid Adom, Syed Muhammad Mamduh, Zaffry Hadi Mohd Juffry, Victor Hernandez Bennetts

**Affiliations:** 1Faculty of Electrical Engineering Technology, Universiti Malaysia Perlis (UniMAP), Arau 02600, Malaysia; abdulnasser994@gmail.com (A.N.A.); latifahmunirah@unimap.edu.my (L.M.K.); abdhamid@unimap.edu.my (A.H.A.); smmamduh@unimap.edu.my (S.M.M.); zaffryhadi@gmail.com (Z.H.M.J.); 2Centre of Excellence for Advanced Sensor Technology (CEASTech), Universiti Malaysia Perlis (UniMAP), Arau 02600, Malaysia; 3B3 Consulting Group, 703 61 Örebro, Sweden; victor.hernandez@b3.se

**Keywords:** MOX sensors, cross-sensitivity, 3D linear regression, temperature, humidity, drift correction

## Abstract

For decades, Metal oxide (MOX) gas sensors have been commercially available and used in various applications such as the Smart City, gas monitoring, and safety due to advantages such as high sensitivity, a high detection range, fast reaction time, and cost-effectiveness. However, several factors affect the sensing ability of MOX gas sensors. This article presents the results of a study on the cross-sensitivity of MOX gas sensors toward ambient temperature and humidity. A gas sensor array consisting of temperature and humidity sensors and four different MOX gas sensors (MiCS-5524, GM-402B, GM-502B, and MiCS-6814) was developed. The sensors were subjected to various relative gas concentrations, temperatures (from 16 °C to 30 °C), and humidity levels (from 75% to 45%), representing a typical indoor environment. The results proved that the gas sensor responses were significantly affected by the temperature and humidity. The increased temperature and humidity levels led to a decreased response for all sensors, except for MiCS-6814, which showed the opposite response. Hence, this work proposed regression models for each sensor, which can correct the gas sensor response drift caused by the ambient temperature and humidity variations. The models were validated, and the standard deviations of the corrected sensor response were found to be 1.66 kΩ, 13.17 kΩ, 29.67 kΩ, and 0.12 kΩ, respectively. These values are much smaller compared to the raw sensor response (i.e., 18.22, 24.33 kΩ, 95.18 kΩ, and 2.99 kΩ), indicating that the model provided a more stable output and minimised the drift. Overall, the results also proved that the models can be used for MOX gas sensors employed in the training process, as well as for other sets of gas sensors.

## 1. Introduction

Metal oxide (MOX) gas sensors have been used in many applications, such as monitoring indoor and outdoor air quality, due to their design simplicity, longer lifetime, high sensitivity, and lower price [1]. Studies have found that the MOX gas sensor has better characteristics than other gas sensor types [2]. However, these sensors suffer from response drift and are strongly affected by ambient conditions, such as temperature and humidity [3,4,5,6,7]. Managing the effect of temperature and humidity on semiconductor electrical conductivity has been a significant challenge for more than 70 years. The high market demand for MOX gas sensors has attracted the attention of researchers to improve their performance [8].

The fundamental principles of MOX gas sensors are based on redox reactions that take place on the surface of the sensitive layer. MOX sensors are highly affected by the temperature of the MOX layer. Conventional operation of conductometric MOX gas sensors takes place on the sensing layer at temperatures in the range of 200–500 °C [9,10,11]. This temperature range is essential for stimulating the relevant chemical reaction and improving its selectivity and rate, usually achieved using a built-in heater resistor. However, the operational temperature requirement limits applications in certain circumstances, such as explosive environments [2].

Moreover, MOX gas sensors suffer from high sensitivity to air humidity [4,12]. Generally, water adsorbed on the sensing surface increases the resistance of the sensing layers and blocks the reaction site, causing gas sensor response drift. However, the effect of humidity on the sensor’s response can be decreased if the temperature is maintained at well above 100 °C [13]. In previous studies, it has been found that the decrease in gas sensitivity is caused by the limited surface and thus leads to a decrease in the activities of chemisorption between the target gases and the MOX layer. In this case, the baseline resistance of the gas sensor is modified [3]. Qi et al. reported that the effect of humidity interference results from the water molecules acting as a barrier against acetylene adsorption [14]. Therefore, the resistance of the water molecules affected the chemisorption of the sensing material.

Several studies have focused on removing or correcting the cross-sensitivity of MOX gas sensors to temperature and humidity. Cross-sensitivity results in response drift because of changes in the surface reaction of MOX gas sensors. Therefore, many researchers have extensively studied the cross-sensitivity of the sensor toward several factors and proposed an alternative to enhance its response. Wang discussed the cross-sensitivity that affects the MOX sensor response, including the chemical components, surface modification, microstructures of sensing layers, and ambient temperature and humidity [15]. Chauhan and Singh developed an optical pH sensor using a titanium-dioxide–silicon-dioxide (TiO_2_–SiO_2_) composite layer to enhance the temperature cross-sensitivity [16]. Ghosh observed the cross-sensitivity of a copper oxide thin film toward several volatile organic compounds (VOC) sensing [17]. Nair conducted a study to improve the zinc oxide (ZnO) sensing response affected by the cross-sensitivity toward humidity by proposing a new doped (ZnO) sensor [3]. Another alternative to enhance the cross-sensitivity is to develop a model that accounts for the specific disturbance [18].

Sensor response drift is one of the most challenging problem in gas sensor technology because the temporal shift of the sensor response is observed under the same working conditions. Such operating condition factors cause phenomena such as dust, speed, wind, and fluctuations of the ambient environmental variables (i.e., humidity and temperature) and system hardware [9,19]. Therefore, the ability of a gas sensor to work for an extended time is limited [20]. Hence, the drift in the sensor response is a significant issue that must be addressed to overcome gas sensor sensitivity and selectivity problems [21]. Furthermore, to enhance the accuracy and reliability of MOX gas sensors, a regression model should be developed to reduce the response drift, mainly owing to temperature and humidity.

The regression method can be classified into univariate and multiple linear regression models. Univariate regression is a model that can be tested for only one independent variable. In contrast, the multiple regression model can describe the relationships between several independent variables and a response variable. Based on studies performed by Badura, raw data were fitted to univariate and multiple regression models [22]. The R^2^ value for the multiple regression analysis was 0.874, which is higher than the value of R^2^ (0.801) obtained from the univariate regression analysis. This shows that the multiple regression analysis fits the data better.

Moreover, in multiple regression analysis, more factors can be considered and form a complex model, such as temperature and humidity. This method has the potential to solve the cross-sensitivity problem. The findings of Sohn were used to create partial-least-squares calibration models [23]. The effects of humidity on the sensor array response and partial-least-squares prediction ability were studied. It was demonstrated that the partial-least-squares model could accurately calibrate the impacts of humidity fluctuations. In a study by Ojha, a linear regression statistical approach was adopted to solve the issue of cross-sensitivity from the simultaneous detection of multiple gases [24]. A linear regression model can be potentially applied to control the drift affected by multiple factors, such as temperature and humidity. In the Kamarudin study, different linear regression models were used to identify the ideal model to improve the drift caused by temperature and humidity [25,26]. Two TGS-2600 sensors were exposed to different ethanol solution concentrations, and drift correction was performed using the proposed model. The results show that the linear regression model could correct the cross-sensitivity of the sensor to ambient temperature and humidity.

In this study, we attempt to learn the relationship between the response of MOX gas sensors towards the ambient temperature and humidity. The novelty of the work is that we proposed new correction models for specific MOX sensors (i.e., MiCS-5524, GM-402B, GM-502B, and MiCS-6814) to minimise the effect of ambient temperature and humidity on its response. Previous studies have failed to develop a model that includes both ambient temperature and humidity parameters in a single model. In addition, some studies are also concerned with the effect of the heater’s temperature rather than the ambient temperature. With this improvement, the gas sensors are expected to provide a more stable response, less affected by environmental conditions fluctuation.

## 2. Experimental

### 2.1. Metal Oxide Gas Sensor

Generally, MOX gas sensors consist of metal oxide semiconductor layers that react to targeted gas based on the conductance principle. A simple electrical circuit can be developed to convert the change in conductivity to an output signal, indicating the gas concentration. A general electrical schematic diagram used to monitor the MOX sensor response is shown in Figure 1, where $R_{S}$ is the sensor resistance, $R_{L}$ is the load resistance, $R_{H}$ is the heater resistance, $V_{C}$ is the reference voltage for the measurement, $V_{H}$ is the voltage heater, and *V_L_* is the voltage across $\;R_{L}$. The sensor resistance $R_{S}$ can be calculated using Equation (1).
(1)$$R_{S}=\frac{V_{C}-V_{L}}{V_{L}}\times R_{L}\;$$

In addition, the characteristics and response of the MOX sensor are highly affected by the temperature of the metal oxide layer. The temperature range of the layer is said to lie between 200 °C and 500 °C [26,27,28] and is achieved using a built-in heater resistor. This temperature range is essential for stimulating the relevant chemical reaction and improving its selectivity and response rate [26]. Moreover, the effect of humidity on the sensor’s response could also be decreased as the temperature is kept well above 100 °C [26].

Table 1 shows four MOX sensors selected for this study and their characteristics. The selection is based on the detection range and targeted gas. These types of sensors were chosen throughout this experiment due to their ability to provide the desired response to the set parameters, temperature, humidity, and gas type, in a way that matches the scope of this study.

### 2.2. Sensor Array Module and Partially Closed Chamber

A sensor array module was developed that consisted of sensor arrays and was covered by a partially closed chamber, as shown in Figure 2 [27,28]. Four sensors were studied in response to ethanol. A temperature sensor (LM35) and humidity sensor (HIH-5030) were also added to measure the ambient temperature and humidity, respectively. A partially closed chamber design was used to make the inflow and outflow of ethanol gas possible. The size of the chamber base was 6 mm high, 62 mm wide, and 84 mm long, while the size of the cover for the sensor array module was 50 mm high, 62 mm wide, and 84 mm long. The use of this chamber eliminated unwanted effects from ambient air. The chamber consisted of airstrips at the top cover that make possible the exchange of the air in the chamber and the air outside the chamber (i.e., from the incubator). This feature enables the ambient temperature and humidity to be controlled using a temperature and humidity incubator. The information on the gas flow was not a concern in this study because it is a fixed parameter, as long as the concentration of gas was uniform throughout the experiment. The assembled sensor array module was tested inside an incubator to verify the functionalities and responses to temperature and humidity changes.

### 2.3. Experimental Setup and Data Collection

Figure 3 shows the hardware setup and a schematic diagram of the experimental setup. The hardware setup contained a measuring system for collecting gas sensor response data at variable temperatures, humidity, and ethanol gas concentrations to model the gas sensor response drift. The setup consisted of a sensor array module, temperature and humidity sensors, a Personal Computer (PC) with LabVIEW software, a data logger (DAQ), pumps, a power supply, an incubator, and a monitor. The gas flow started with the inlet pump feeding air into a carbon filter. A carbon filter was added in this experiment to clean the intake air and filter out VOCs that may be present and affect the experiment before the air was fed to the bubbler of the ethanol solution. The conical flask contained prepared ethanol solutions of different concentrations to produce varying gas concentrations during the bubbling process. The gas was then fed into the partially closed chamber, which contains the sensor array module (see Figure 2). The partially closed chamber would allow the exchange of gas in the incubator with the air inside the chamber to control the temperature and humidity. Finally, the outlet pump drained the gas to ensure a continuous flow. The outlet pipe was placed slightly above and in between the gas sensors in the array, ensuring that all sensors were exposed to the same gas concentration.

In order to obtain different relative concentrations of VOC, six different ethanol solutions were prepared (i.e., 0%, 0.05%, 0.2%, 0.5%, 1%, and 2%) by mixing an ethanol solution with certain volumes of distilled water. These solutions were labelled as c0, c50, c200, c500, c1000, and c2000 respectively, and bubbled with clean air in different experimental settings. Note that the actual concentration was not measured since the main objective is to study the variation of the sensor’s response with respect to varying temperatures and humidity at relatively different levels of gas concentrations. Figure 4 shows the temperature and humidity controls for the experiment regarding time, where the total experiment duration for each concentration level is 630 min. The responses of the gas, temperature, and humidity sensors were acquired every second throughout the experiment. In the first 15 min, the outlet pump was turned on to purge the sensor chamber while enabling the incubator to adjust its internal air temperature to 16 °C with 75% humidity. Then, the inlet pump was switched on to enable the ethanol gas to enter the chamber. The process was performed for 15 min until gas equilibrium was achieved.

The incubator was set to cause the temperature to oscillate between 16 °C and 30 °C uniformly between the 30th and 630th min, while the humidity was gradually reduced from 75% to 45% and increased again to 75% within the same period. This setting was used to minimise the time effect. It is because, over the continuous bubbling process, the concentration of ethanol gas might decrease. The calculated average of the outcomes of both temperature controls (ramp up and ramp down) can generate a more reliable pattern. Finally, the incubator was set to ramp up the temperature to 20 °C and ramp down the humidity to 65%, leaving the incubator to meet these conditions for 15 min, and then it was left for 30 min. The gas sensor responses to these conditions during the last 30 min were used as reference data (i.e., target data for linear regression), while the period between the 30th and 630th min was used as training data. The whole experiment is repeated with all humidity levels and at different ethanol solution concentrations.

### 2.4. Data Analysis

In this study, a gas sensor response model was created. Given $R_{S}$, temperature, and humidity values, the model *F* can predict the corrected response $R_{S}^{*}$ that eliminates the drift resulting from temperature and humidity variations. This function is described in Equation (2),
(2)$$R_{s}^{*}=F(R_{s},T,H)\;$$
where $R_{s}$ is the sensor response in kΩ, T is the ambient temperature in °C, and H is the relative humidity in percentage. Because the corresponding target value was provided, this modelling process was considered as a supervised learning problem. Different models have been proposed and tested to select the ideal model. Models with varying terms of interaction were generated using a polynomial function to model the drift of the sensor. In other words, seven types of models (m1 to m7) were developed, each consisting of different interaction terms (i.e., a combination of $R_{s}$, T, and H). Mathematically, they can be written as Equations (3)–(9),
(3)$$m1:\;F\left(R_{s},\;T,\;H\right)=C_{0}+C_{1}R_{s}+C_{2}T+C_{3}H\;$$
(4)$$m2:\;F\left(R_{s},\;T,\;H\right)=C_{0}+C_{1}R_{s}+C_{2}R_{s}TH\;$$
(5)$$m3:\;F\left(R_{s},\;T,\;H\right)=C_{0}+C_{1}R_{s}+C_{2}R_{s}T+C_{3}R_{S}H\;\;\;$$
(6)$$m4:\;F\left(R_{s},\;T,\;H\right)=C_{0}+C_{1}R_{s}+C_{2}T+C_{3}H+C_{4}R_{S}TH\;$$
(7)$$m5:\;F\left(R_{s},\;T,\;H\right)=C_{0}+C_{1}R_{s}+C_{2}T+C_{3}H+C_{4}R_{S}T+C_{5}R_{S}H\;$$
(8)$$m6:\;F\left(R_{s},\;T,\;H\right)=C_{0}+C_{1}R_{s}+C_{2}T+C_{3}H+C_{4}R_{S}T+C_{5}R_{S}H+C_{6}TH\;$$
(9)$$m7:\;F\left(R_{s},\;T,\;H\right)=C_{0}+C_{1}R_{s}+C_{2}T+C_{3}H+C_{4}R_{S}T+C_{5}R_{S}H+C_{6}TH+C_{7}R_{s}TH\;$$
where $C_{0}-C_{n\;}$ are the polynomial coefficients. The values of the coefficients were obtained through linear regression to minimise the sum of the squares of the errors. A 3D linear regression model was used to observe the performance of the ideal model used for the gas sensor response drift correction. A mathematical model was independently developed based on the selected basis function for each gas sensor and based on the data collected.

The k-fold cross-validation (CV) approach was utilised to evaluate the performance and provide a measure of fit, i.e., mean square error (MSE), to select candidate models for the dataset. The dataset was randomly divided into k equally sized subsamples in this approach. At every sample time, k − 1 subsample was gathered to form the training dataset, and the other subsample was used as the validation dataset. The procedure was repeated k times so that every k subsample was in a test set only once. Then, the entire k-fold process was repeated n times, with new randomly drawn subsamples. Finally, the MSEs were obtained from all the n repetitions, the k-folds were averaged, and the standard deviation was calculated. The technique has an advantage over the conventional testing method (e.g., dividing data into 70% for the training set and 30% for the test set) in that all samples are used for both training and testing. Therefore, more reliable results were obtained.

## 3. Results and Discussion

### 3.1. Effect of Temperature and Humidity on Gas Sensor Response

In this section, the plots of the sensors’ responses acquired based on the experimental setup described in Section 2.3 are presented. Two plot types (2D and 3D) were used to observe the gas sensor response. Figure 5 illustrates the pattern of the measured ambient temperature and humidity at ethanol solution concentrations of 0%, 0.05%, 0.2%, 0.5%, 1%, and 2%, which are indicated by c0, c50, c200, c500, c1000, and c2000, respectively. The temperature was controlled in the range of 16 °C to 30 °C, but the actual temperature was found to be between 20 °C and 35 °C because of the heat emitted from the sensor heater and mixing process. The humidity gradually decreased from 75% to 45% and then increased to 75%. In addition, the humidity plot had many spikes because the incubator had difficulty in controlling the level, which is leading to fast fluctuations at higher temperatures. Nevertheless, data filtering was not needed because the sensor response changes according to the spike and such an effort may significantly influence the reliability and accuracy of the data.

The overall patterns of the sensor responses with respect to varying temperature and humidity were analysed and used to predict the general form of the Equation used for modelling. Figure 6 shows 2D plots of sensor responses for MiCS-5524, GM-402B, GM-502B, and MiCS-6814 against time at different temperatures, humidity, and concentrations of the ethanol solution. The sensors behaved similarly, and a lower gas sensor response was observed at higher concentrations of ethanol solution. In addition, the sensor responses fluctuated with the temperature pattern but had the opposite value. The effect of humidity on the sensor responses was also reversed, but it was less significant than the temperature. All the observations are valid for all the sensors tested except MiCS-6814, which exhibited the opposite pattern.

Figure 7, Figure 8, Figure 9 and Figure 10 show the 3D surface plots for all gas sensor responses. The mean responses with respect to temperature and humidity were also calculated and plotted. These plots show the relationship between the sensor response, temperature, and humidity. The 3D surface plots reveal that the peak sensor response is usually obtained when the temperature and humidity are at their lowest. The sensor response is the lowest when the two parameters are highest.

Generally, based on the analysis of the 2D and 3D plots for each gas sensor response, two main observations can be made:The sensor responses decrease almost linearly with increasing temperature.The sensor responses decrease almost linearly with increasing humidity.

Hence, the models proposed to correct *R_S_* values resulting from temperature and humidity variations are examined through a multiple linear regression model in the next section. The correction of gas response fluctuations was studied and is valid for the examined VOCs.

### 3.2. Correction Model for Sensor Drift Caused by Ambient Temperature and Humidity

The linear regression method with 10-fold CV was utilised to show the effect of temperature and humidity fluctuations on the gas sensor response, and the correction was performed using supervised linear regression. First, the k-fold CV technique was utilised to measure the fit (i.e., mean square errors) of the candidate models. The averaged MSEs and standard deviations for all models (see Section 2.4) when using 10-fold CV with 100 iterations are shown in Figure 11. Seven models (m1 to m7) were tested, each consisting of different interaction terms (i.e., a combination of *R_S_*, *T*, and *H*). In general, the MSE patterns for all sensors decreased as the complexity of the model increased (i.e., from m1 to m7). The MSEs for m1 and m2 for all sensors were significantly higher than for the other models because of the absence of important interaction terms (i.e., interactions between *R_S_*, temperature, and humidity, which thus should be included in the model). The averaged MSE dropped substantially from m2 to m3, suggesting that the terms T and H should be separated. The findings also indicate that the individual temperature and humidity terms are significant, as in m4. The change from C4RsTH in m4 to (*C_4_R_S_T* + *C_5_R_S_H*) in m5 also improved the result.

The results show similar average MSEs and standard deviations in m5, m6, and m7. Therefore, these can potentially be used as a model to solve this problem. However, m5 was chosen because of its lower complexity and avoiding overfitting due to additional interaction terms. In addition, the interaction between the temperature and humidity (*TH* or *R_S_TH*) in m6 and m7 on the sensor response has never been discussed in the literature. With this interaction term, a false correction may be produced.

Second, the models were independently generated based on the m5 basis function and utilised all gas sensors’ entire dataset. Third, the coefficients for model m5 were determined based on the supervised linear regression method explained in Section 2.4. The models generated for all gas sensors are given in Equations (10)–(13), respectively. The model works such that, given the values of *R_S_*, temperature, and humidity, it can predict the corrected *R_S_* value (denoted by *R_S_**) that minimises the effects of temperature and humidity.
(10)$$F_{MiCS5524}\left(R_{S\_5524},T,H\right)=-7.4941-0.6914R_{S\_5524}-0.0531T+0.2860H+0.0484R_{S\_5524}T+0.0071R_{S\_5524}H\;$$
(11)$$F_{GM402B}\left(R_{S\_402},T,H\right)=-10.4272+0.0978R_{S\_402}+0.0128T+0.3387H+0.0111R_{S\_402}T+0.0088R_{S\_402}H\;$$
(12)$$F_{GM502B}\left(R_{S\_502},T,H\right)=-48.4869+0.0135R_{S\_502}+2.5624T+0.9459H+0.0251R_{S\_502}T+0.0027R_{S\_502}H\;$$
(13)$$F_{MiCS6814}\left(R_{S\_6814},T,H\right)=10.2243+0.7495R_{S\_6814}-0.1953T+0.0672H+0.0016R_{S\_6814}T+0.0081R_{S\_6814}H\;$$

Figure 12, Figure 13, Figure 14 and Figure 15 show 3D surface plots of the corrected response $R_{S}^{*}$ for MiCS-5524, GM-402B, GM-502B, and MiCS-6814. The mean corrected responses with respect to temperature and humidity were also plotted to provide a clearer response pattern based on the temperature and humidity relation. The plots for $R_{S\_MiCS5524}^{*}$, $R_{S\_GM402B}^{*},R_{S\_GM502B}^{*}$, and $R_{S\_MiCS6814}^{*}\;$ are flatter at varying temperatures and humidity than the plots in Figure 7, Figure 8, Figure 9 and Figure 10. This proves that the model can minimise the effects of temperature and humidity on the gas sensor response.

Finally, the reliability of the proposed models (i.e., Equations (10)–(13)) was further verified by using different gas sensors instead of the ones used in the model generation experiments. To avoid confusion, these gas sensors were denoted as MiCS-5524V2, GM-402BV2, GM-502BV2, and MiCS-6814V2. The experiment was conducted in a closed room (closed door and window), and an air conditioner was used to control the temperature and humidity. The sensors were exposed directly to ambient air instead of being enclosed in the previously mentioned chamber. Figure 16 shows the resulting plots and the different times the air conditioner setting was changed.

To observe the gas sensor response at different temperatures, the setting of the air conditioner was repeatedly changed between 30 °C and 16 °C for two cycles with an interval of 2 h between sets. However, the room’s actual temperature did not reach the set values (30 °C and 16 °C) because of the air conditioner efficiency and ambient conditions outside the room. Furthermore, the narrow openings of the windows and door result in some heat transfer and minor air exchange. In addition, the humidity throughout the experiments depended on the air conditioner’s setting. Changes in temperature and humidity were observed to affect all the gas sensor responses. As shown in the plots, the gas sensor responses were inversely proportional to the change in temperature and humidity, except for MiCS-6814. Therefore, the modelled gas sensor response could minimise the effects of temperature and humidity. However, the plots slowly deviated from the reference over time because of changes in air composition throughout the experiment. This was more pronounced when the air conditioner setting was changed, owing to significant air exchange, while the door was opened to change the setting. Based on the results, the general model was verified, and the cross-sensitivity affected by temperature and humidity was minimised for all sensors, even though different sensors were used. Moreover, the experiments revealed that the model was effective when the sensor was directly exposed to the ambient environment.

Table 2 summarises the mean and standard deviation of the graphs shown in Figure 16, where the number of samples is 28,800. The results show a slight difference between the mean of the measured data and the corrected data. In addition, for the comparison between the standard deviations, the corrected data show a smaller standard deviation value than the measured data. The smaller standard deviation values suggest less fluctuation in the gas sensor response. This means that the model effectively minimised the effects of ambient temperature and humidity. Among the sensors, MiCS-6814V2 exhibited the lowest standard deviation of the corrected data, which indicates that the gas sensor drift was corrected. On the other hand, sensors GM-402BV2 and GM-502V2 had larger standard deviations of corrected data because the mean values are in the range of hundreds. Overall, the standard deviation of the corrected data is seen to be much lower than the standard deviation of the measured data, proving that this model can minimise the effects of ambient temperature and humidity for all gas sensors used in the experiment, as well as other sets of MOX gas sensors.

## 4. Conclusions

MOX gas sensors are widely used in gas-sensing technologies. However, this sensor type is cross-sensitive to the temperature and humidity of the ambient air. Therefore, an attempt was made to correct the MOX gas sensor response for a more reliable gas-sensing operation. The incubator was used to vary the ambient temperature and relative humidity while exposing the gas sensors to six different concentrations of ethanol gas. The results indicate that the response of the sensors decreases almost linearly as temperature and humidity increase, except for the MiCS-6814 sensor, which has the opposite behaviour because of its layer built to react with NH_3_ gas. Therefore, linear regression was performed to produce different models: F_MiCS5524_, F_GM402B_, F_GM502B_, and F_MiCS6814_. The model works such that, given the sensor response (*R_S_*), temperature (*T*), and humidity (*H*), it can predict a corrected sensor response $\left(R_{S}^{*}\right)$ that minimises the temperature and humidity effects. The model of each sensor was tested using data from other sensors for model verification. The experiment was conducted in a closed room with different air conditioner settings (16–30 °C). Although the temperature and humidity changed throughout the experiment, the corrected sensor response was more uniform and stable than the initial response. This finding is supported by the smaller standard deviation of the corrected data (i.e., 1.66 kΩ, 13.17 kΩ, 29.67 kΩ, and 0.12 kΩ for MiCS-5524, GM-402B, GM-502B, and MiCS-6814, respectively) compared with the measured data. Overall, the results also proved that the models can be used for the MOX gas sensors employed in the training process and other sets of gas sensors.

## Figures and Tables

**Figure 1 sensors-22-03301-f001:**
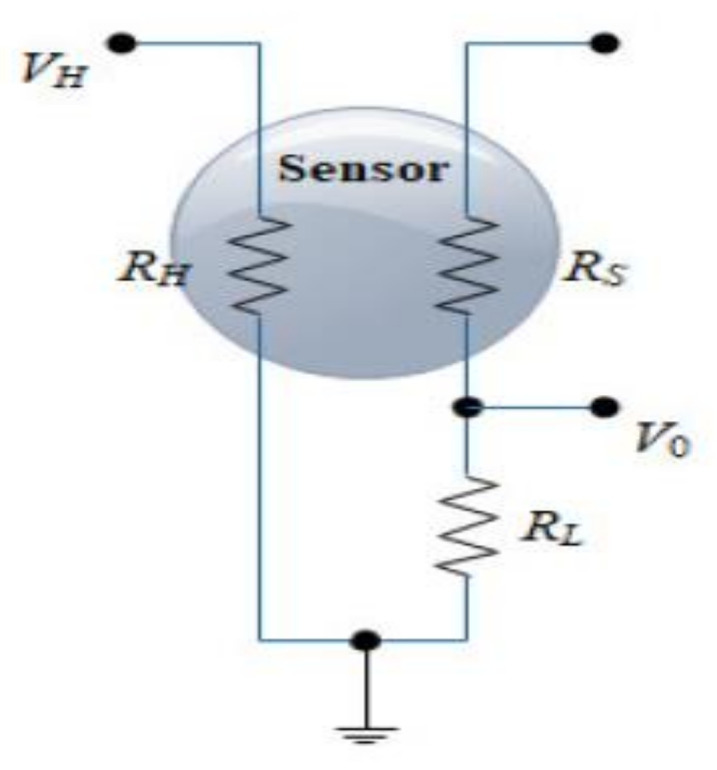
The electronic circuit of the MOX gas sensor is used for monitoring the sensor response.

**Figure 2 sensors-22-03301-f002:**
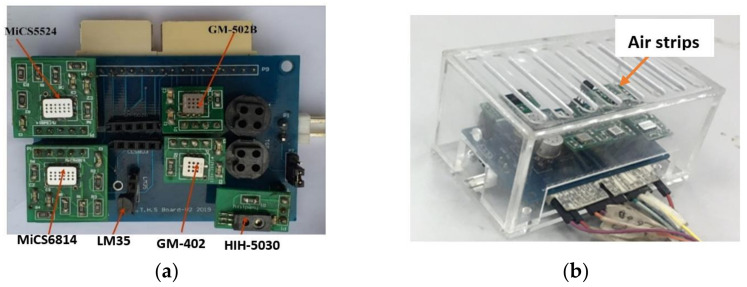
(**a**) Gas sensor modules developed; (**b**) fabricated partially closed chamber using acrylic with all gas sensor modules.

**Figure 3 sensors-22-03301-f003:**
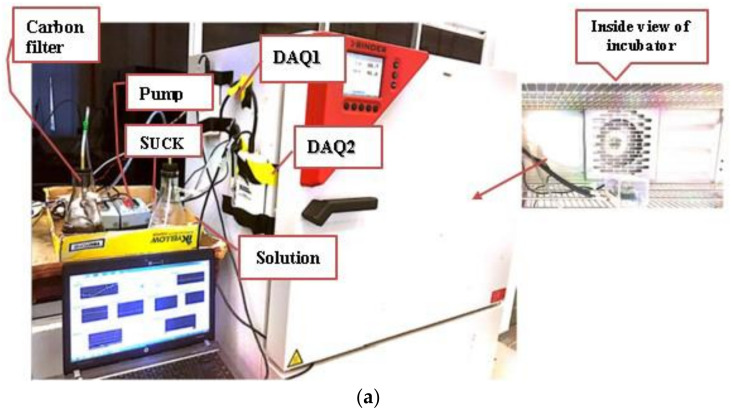
(**a**) Picture of hardware setup at incubator control; (**b**) schematic diagram for the experimental setup.

**Figure 4 sensors-22-03301-f004:**
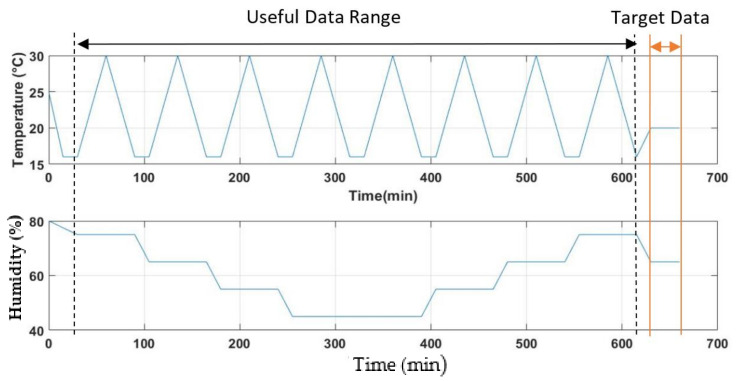
Temperature and humidity control of the incubator: Useful data range and target data range of 30th to 615th and 630th to 660th min, respectively.

**Figure 5 sensors-22-03301-f005:**
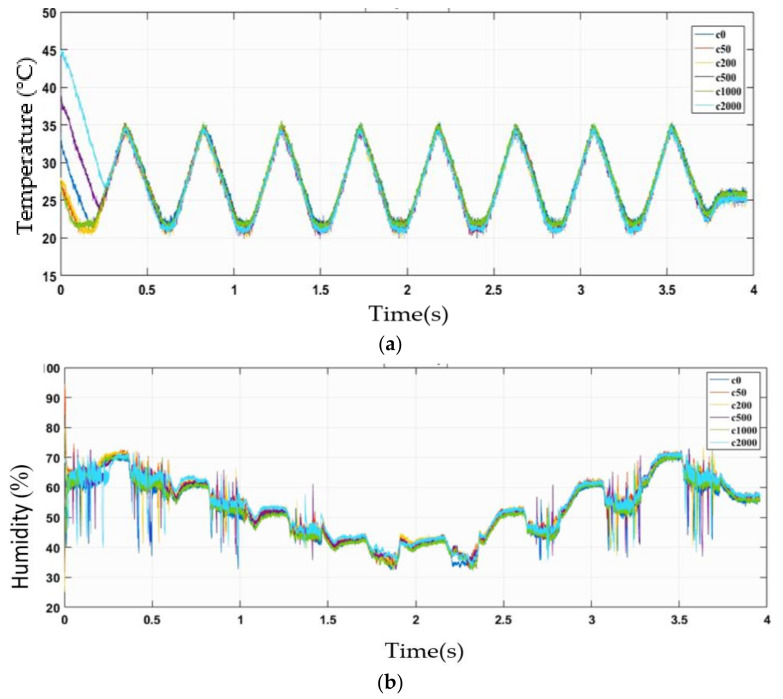
Two-dimensional plot of results using ethanol solutions of c0, c50, c200, c500, c1000, and c2000 concentration and humidity settings of h75, h65, h55, h45, h45, h55, h65, and h75: (**a**) temperature setting and (**b**) humidity setting.

**Figure 6 sensors-22-03301-f006:**
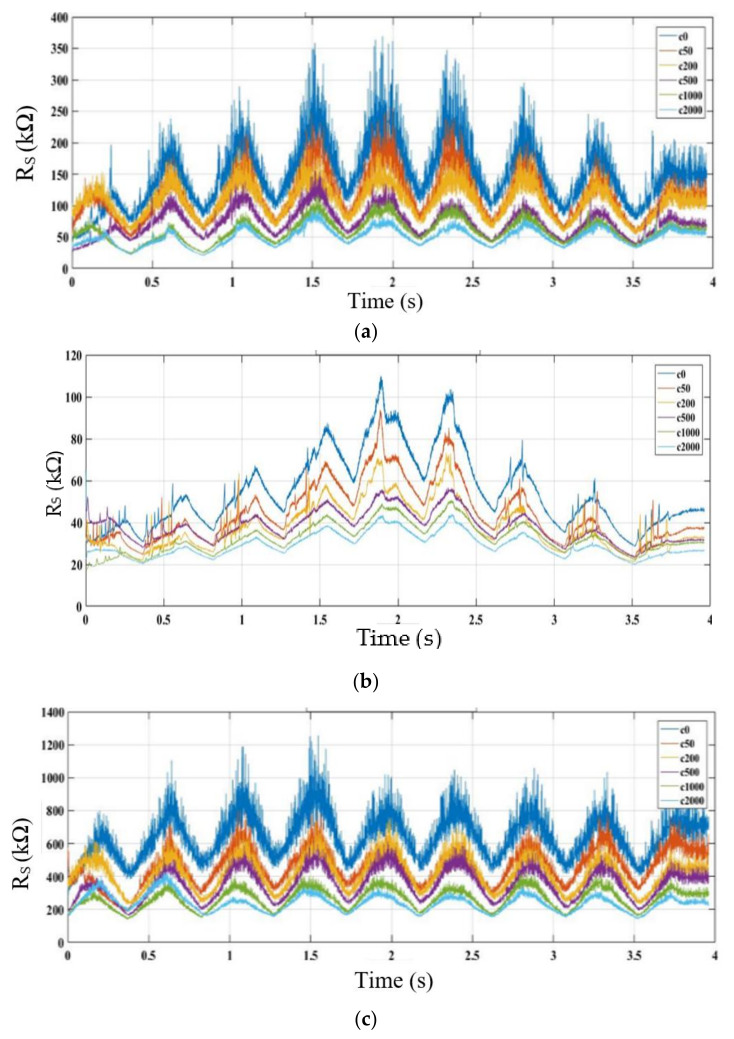
Two-dimensional plots of gas sensor response *R_S_* against time at different temperatures, humidity, and concentration levels of ethanol solution (c0, c50, c200, c500, c1000, and c2000): (**a**) MiCS-5524, (**b**) GM-402B, (**c**) GM-502B, and (**d**) MiCS-6814.

**Figure 7 sensors-22-03301-f007:**
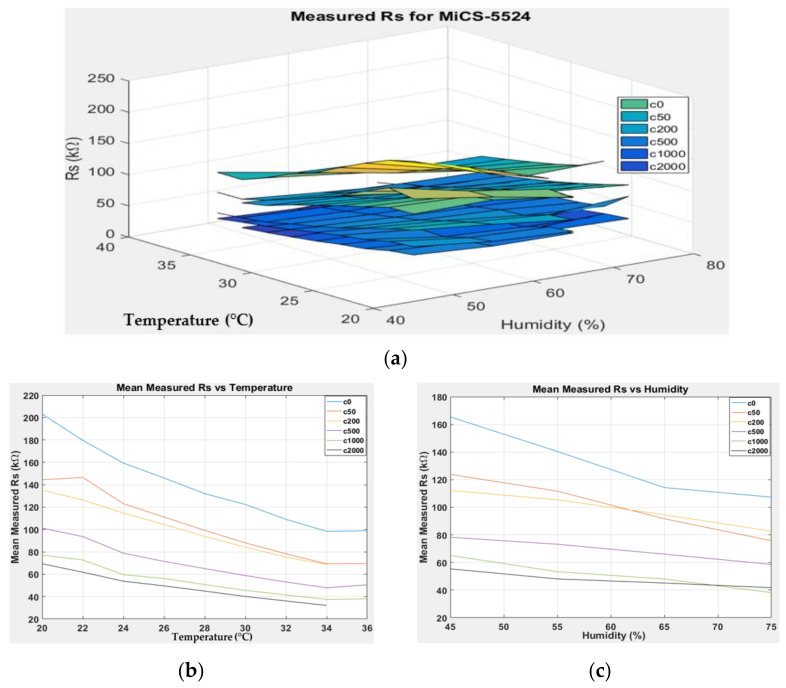
(**a**) Three-dimensional surface graph of MiCS-5524 measured *R_S_* at varying temperature and humidity levels and different ethanol solution concentrations, (**b**) mean of measured *R_S_* versus temperature, and (**c**) mean of measured *R_S_* versus humidity.

**Figure 8 sensors-22-03301-f008:**
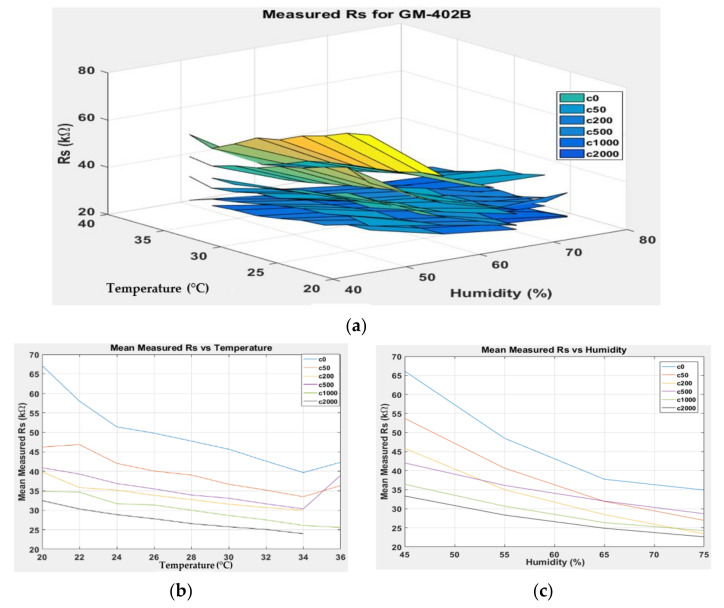
(**a**) Three-dimensional surface graph of GM-402B measured *R_S_* at varying temperature and humidity levels and different ethanol solution concentrations, (**b**) mean of measured *R_S_* versus temperature, and (**c**) mean of measured *R_S_* versus humidity.

**Figure 9 sensors-22-03301-f009:**
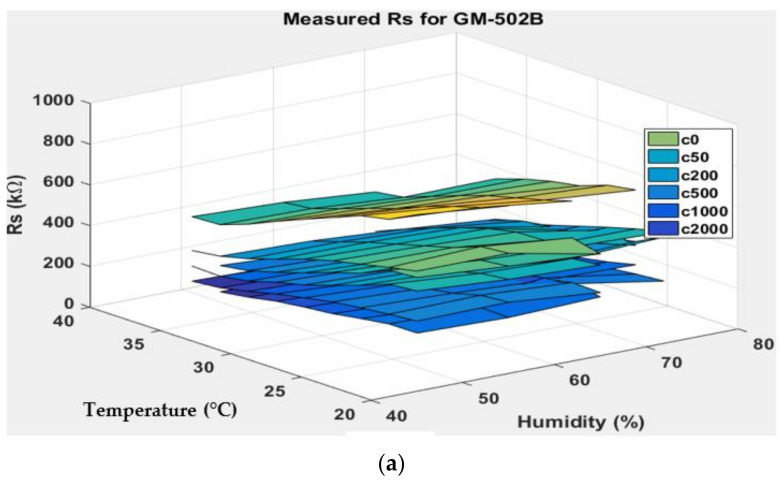
(**a**) Three-dimensional surface graph of GM-502B measured *R_S_* at varying temperature and humidity levels and different ethanol solution concentrations, (**b**) mean of measured *R_S_* versus temperature, and (**c**) mean of measured *R_S_* versus humidity.

**Figure 10 sensors-22-03301-f010:**
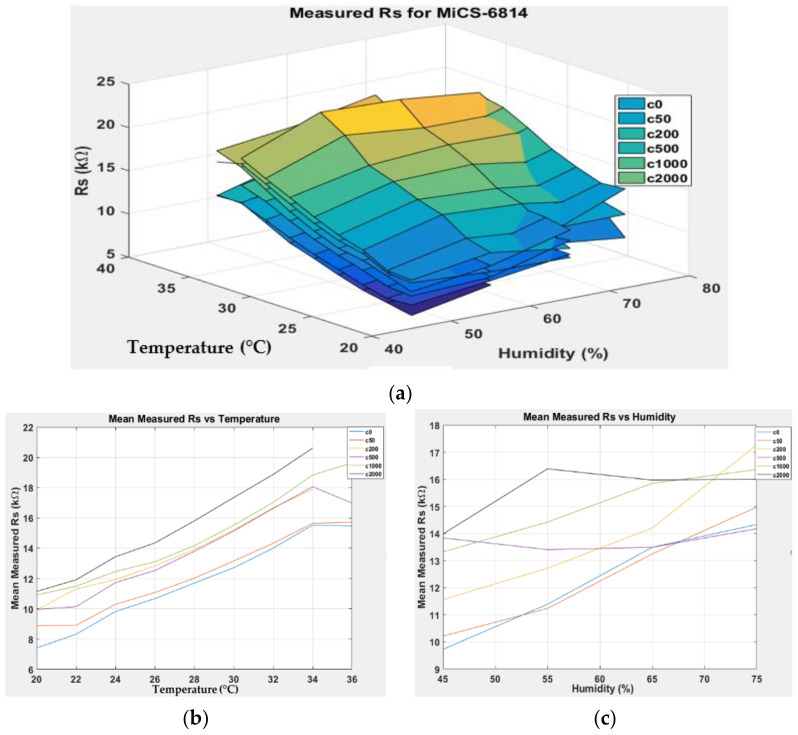
(**a**) Three-dimensional surface graph of MiCS-6914 measured *R_S_* at varying temperature and humidity levels and different ethanol solution concentrations, (**b**) mean of measured *R_S_* versus temperature, and (**c**) mean of measured *R_S_* versus humidity.

**Figure 11 sensors-22-03301-f011:**
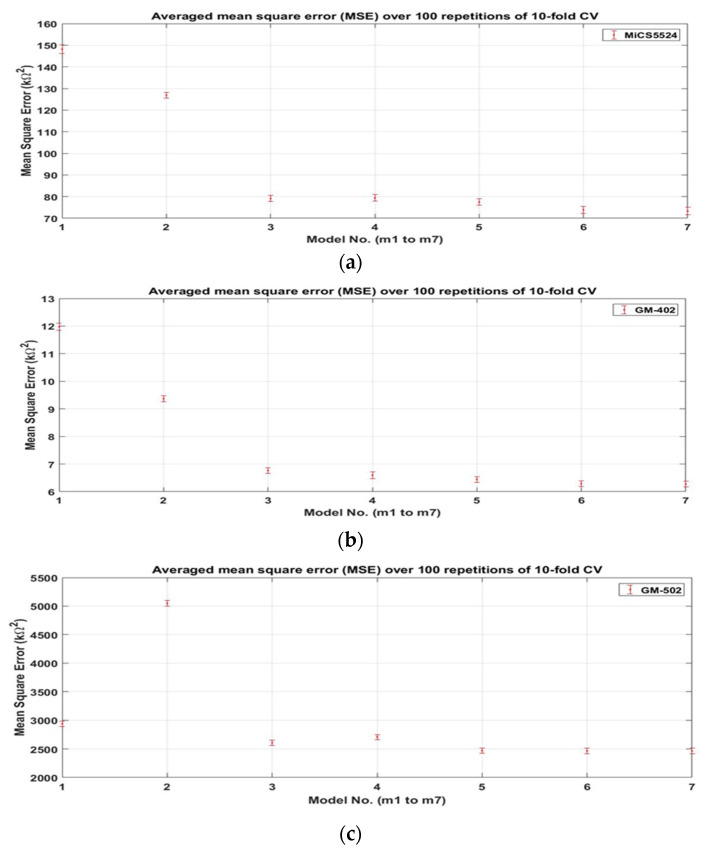
Averaged MSEs and standard deviation for all gas sensors using 10-fold CV with 100 repetitions to find the optimal model selection from m1 to m7: (**a**) MiCS-5524, (**b**) GM-402, (**c**) GM-502B, and (**d**) MiCS-6814.

**Figure 12 sensors-22-03301-f012:**
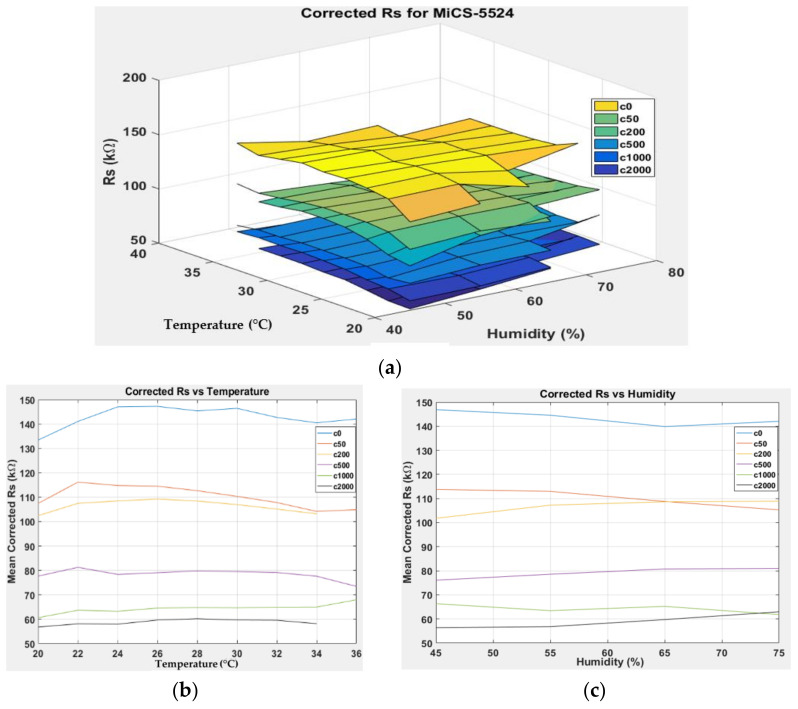
(**a**) Three-dimensional surface graph of MiCS-5524 corrected *R_S_* using m5 model at varying temperature and humidity levels and different ethanol solution concentrations, (**b**) mean of corrected *R_S_* versus temperature, and (**c**) mean of corrected *R_S_* versus humidity.

**Figure 13 sensors-22-03301-f013:**
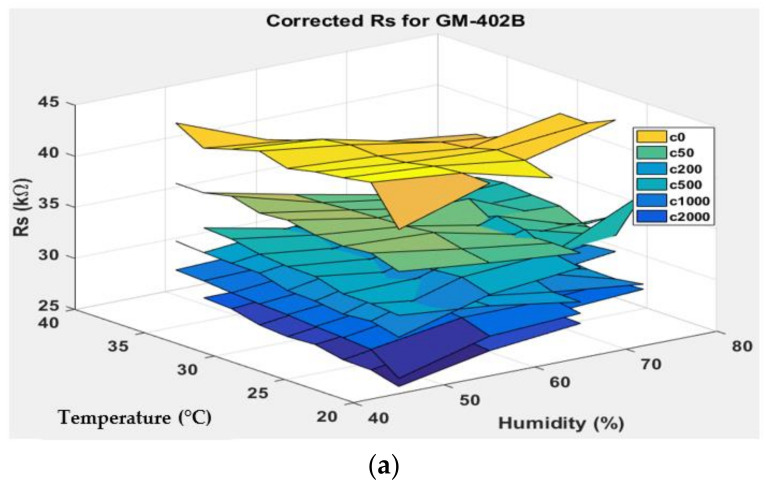
(**a**) Three-dimensional surface graph of GM-402B corrected *R_S_* using m5 model at varying temperature and humidity levels and different ethanol solution concentrations, (**b**) mean of corrected *R_S_* versus temperature, and (**c**) mean of corrected *R_S_* versus humidity.

**Figure 14 sensors-22-03301-f014:**
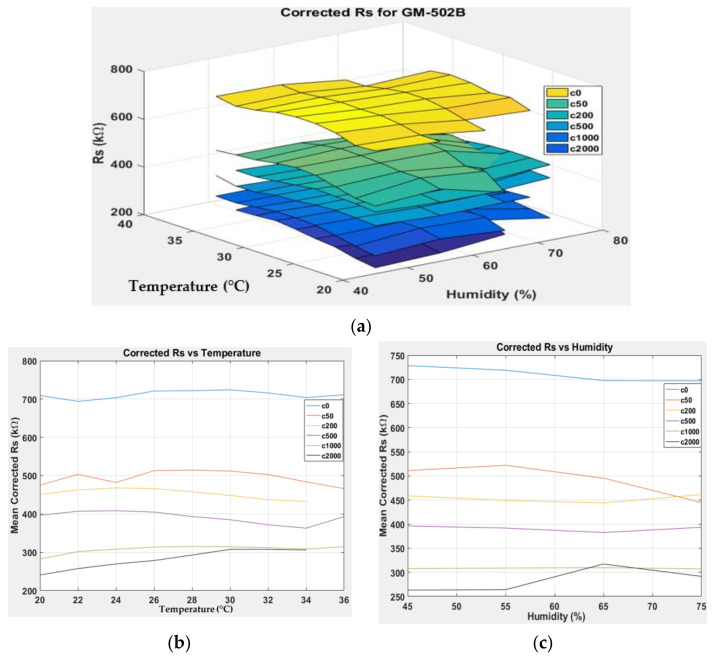
(**a**) Three-dimensional surface graph of GM-502B corrected *R_S_* using m5 model at varying temperature and humidity levels and different ethanol solution concentrations, (**b**) mean of corrected *R_S_* versus temperature, and (**c**) mean of corrected *R_S_* versus humidity.

**Figure 15 sensors-22-03301-f015:**
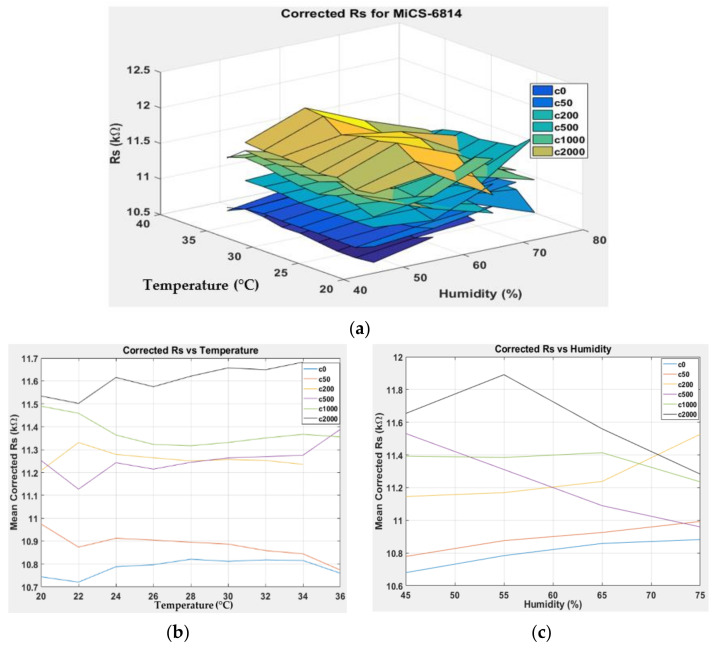
(**a**) Three-dimensional surface graph of MiCS-6814 corrected *R_S_* using m5 model at varying temperature and humidity levels and different ethanol solution concentrations, (**b**) mean of corrected *R_S_* versus temperature, and (**c**) mean of corrected *R_S_* versus humidity.

**Figure 16 sensors-22-03301-f016:**
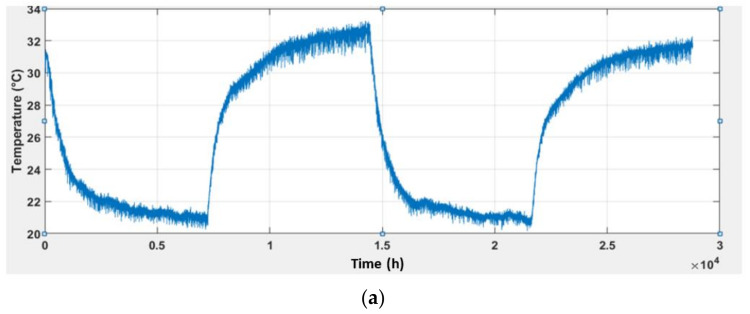
Temperature and humidity controlled using air conditioner: (**a**) Temperature (LM35); (**b**) humidity (HIH-5030); as well as measured and modelled sensor responses inside closed room using (**c**) MiCS-5524V2; (**d**) GM-402BV2; (**e**) GM-502BV2; and (**f**) MiCS-6814V2.

**Table 1 sensors-22-03301-t001:** Type of gas sensors for the PCB board.

Sensor Type	Target Gases	Detection Range	Features
MiCS-5524 [29]	Carbon monoxide Ethanol Hydrogen Methane	1–1000 ppm 10–500 ppm 1–1000 ppm >1000 ppm	Smallest footprint for compact design. Robust MEMS sensor for harsh environments. High-volume manufacturing for low-cost applications.
GM-402B [30]	Methane Propane	1–1000 ppm 1–5000 ppm	Low power consumption. High sensitivity. Fast response. Simple drive circuit.
GM-502B [31]	Carbon monoxide Nitrogen dioxide Ethanol Hydrogen Propane Methane	1–1000 ppm 0.005–10 ppm 10–500 ppm 1–1000 ppm >1000 ppm >1000 ppm	Low power consumption. High sensitivity. Fast response. Simple drive circuit.
MiCS-6814 [32]	Carbon monoxide Ethanol Hydrogen Methane Propane	1–1000 ppm 10–500 ppm 1–1000 ppm 1–500 ppm >1000 ppm	Smallest footprint for compact design. Robust MEMS sensor for harsh environments. High-volume manufacturing for low-cost applications.

**Table 2 sensors-22-03301-t002:** Comparison of mean and standard deviation for measured and corrected data.

Sensor	Mean of Measured Data (kΩ)	Standard Deviation of Measured Data (kΩ)	Mean of Corrected Data (kΩ)	Standard Deviation of Corrected Data (kΩ)
MiCS-5524V2	72.59	18.22	79.30	1.66
GM-402BV2	190.71	24.33	189.29	13.17
GM-502V2	528.92	95.18	515.77	29.67
MiCS-6814V2	13.26	2.99	13.00	0.12

## Data Availability

Not applicable.

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
