# Peer review of "Correction Model for Metal Oxide Sensor Drift Caused by Ambient Temperature and Humidity"

_sensors, 2022, doi:10.3390/s22093301_

Round 1
Reviewer 1 Report
The article deals with the analysis of the behavior of metal oxide gas sensors subjected to the effect of ambient temperature and humidity.
The following issues need to be addressed:
- Abstract
- I suggest reconsidering the abstract by improving the readability and illustrating only the main concepts.
- Introduction
- Lines 102-106 - Sufficient detail are not reported for the ANNs employed for the topic of the paper.
- The novelty of the work should be highlighted.
- Experimental investigation
- The investigated sensors should be better described, reporting their main characteristics.
- Fig2 – What do ‘temperature’ and ‘humidity’ mean?
- Further details should be given about the setup. All information useful for replicating the experiments should be provided.
- Fig3 – The fig. is distorted.
- Lines 176-181 – Improve the clarity of the sentences.
- Fig4 – I do not understand what the time shown in fig. Is it after the gas equilibrium?
- Lines 206-207 – Replace kiloohms with kΩ and degrees Celsius with °C.
- Results
- Figs 5, 6, 7, 8, 9, 10, 11, 12, 13, 14, 15, 16 - The labels are difficult to read. Fit the figures to the page. Moreover, use the correct symbols for the units on the axis (e.g . kΩ and °C).
- Conclusion
- Improve the conclusion section by reporting quantitative data.
Author Response
Dear reviewer, thank you for your precious time and suggestions for improvement. Please find our respond to your concerns in the attachment.

Reviewer 2 Report
1) The motivation for this article is not clear based on the Introduction section. From line 89-114, the authors cite several papers which detail how different modelling techniques have been used by previous researchers to address/model different problems such as drift due to temperature and humidity and cross sensitivity in MOX sensors. The introduction section fails to justify the originality/novelty of this paper. What is new about this article compared to previous literature?
2) Equation 1 in section 2.1 does not seem accurate. Please see previous published literature such as that listed below and justify.
a) García-Orellana, Carlos J., et al. "Low-power and low-cost environmental IoT electronic nose using initial action period measurements." Sensors 19.14 (2019): 3183. (Figure 5, Eqn 4)
b) Hernandez Bennetts, Victor, et al. "Mobile robots for localizing gas emission sources on landfill sites: is bio-inspiration the way to go?." Frontiers in neuroengineering 4 (2012): 20. (Figure 3, Eqn 1)
3) The experimental details in this manuscript are not very clear and it seems control is lacking in data collection. e.g. In line 145-147, the authors note that the inlet air flow is not monitored as it is a fixed value, and the different concentration of ethanol vapor were obtained by bubbling through diluted ethanol solution of varying concentration. (line 158)
i) How was the diluted ethanol concentration solution obtained? Is that ethanol mixed with distilled water? If yes, then doesn't bubbling through the solution introduce additional humidity into the chamber?
ii) If the flows are held constant and the ethanol solution is diluted, doesn't the concentration of the solution and hence ethanol vapor change over the duration of the experiment? It is unclear to me why the authors choose specific ethanol concentration solutions (line 158) but do not control the said concentrations.
Constant vapor concentration can be obtained using N2 flows bubbling through a solvent solution. See previous literature -
Likhite, Rugved, et al. "VOC sensing using batch-fabricated temperature compensated self-leveling microstructures." Sensors and Actuators B: Chemical 311 (2020): 127817. (Eqn 4)
4) The authors use 2 reference sensors (LM35 and HIH5030) for temperature and humidity monitoring. Like all sensors, these reference devices would also suffer from cross sensitivity. The authors should comment on the interaction (if any) of their reference/control sensors with ethanol/humidity/temperature.
5) In lines 406-408, the authors provide the conclusion of their article, but how does the developed model compare against the ones in previous literature? Comments on comparison of this model to existing publications should be added.
Author Response

(The authors gave the same response as above.)

Reviewer 3 Report
The article presents the results of improving the performance of four gas sensors that are produced by the industry and are widely used in environmental monitoring devices. In particular, a model has been developed for correcting the response of sensors, leveling the influence of humidity and ambient temperature. The results obtained are of practical interest and are suitable for publication in Sensors. However, there are several shortcomings that are not of a fundamental nature. However, they should be corrected.
- Lline 128. Replace symbol R_S with ?s ;
- Line 131. The value of ?h is not in equation (1);
- The time scale in figures 5, 6 and 7 should be presented in the same units;
- Fig. 16. The factor 104 seems to be unnecessary on the time scale.
- Lines 332-333. Figures 9-12. Maybe figures 7-10?
Author Response

(The authors gave the same response as above.)

Round 2
Reviewer 1 Report
All my comments have been sufficiently addressed.
However, Fig. 16 continues to report kOhm e hour instead of kΩ and h.
Moreover, the standard deviation is not in kΩ2. In fact, the quantities in the abstract and in tab. 2 are variance.
Author Response
Dear reviewer, thank you for your precious time and suggestions for the improvement of manuscript.

Reviewer 2 Report
Thank you for addressing the comments. I recommend this manuscript for publication.
Author Response

(The authors gave the same response as above.)
